# Nurses’ Roles, Responsibilities and Actions in the Hospital Discharge Process of Older Adults with Health and Social Care Needs in Three Nordic Cities: A Vignette Study

**DOI:** 10.3390/ijerph20196809

**Published:** 2023-09-22

**Authors:** Ann E. M. Liljas, Natasja K. Jensen, Jutta Pulkki, Janne Agerholm

**Affiliations:** 1Department of Global Public Health, Karolinska Institutet, 171 77 Stockholm, Sweden; 2Department of Public Health, University of Copenhagen, 1123 Copenhagen, Denmark; naje@sund.ku.dk; 3The Health Sciences Unit, Faculty of Social Sciences, Tampere University, 33520 Tampere, Finland; 4Department of Neurobiology, Care Sciences and Society, Karolinska Institutet, 171 77 Stockholm, Sweden; janne.agerholm@ki.se

**Keywords:** aging, older people, hospital discharge, healthcare, home care, social care

## Abstract

The hospital discharge process of older adults in need of both medical and social care post hospitalisation requires the involvement of nurses at multiple levels across the different phases. This study aims to examine and compare what roles, responsibilities and actions nurses take in the hospital discharge process of older adults with complex care needs in three Nordic cities: Copenhagen (Denmark), Stockholm (Sweden) and Tampere (Finland). A vignette-based interview study consisting of three cases was conducted face-to-face with nurses in Copenhagen (*n* = 11), Stockholm (*n* = 16) and Tampere (*n* = 8). The vignettes represented older patients with medical conditions, cognitive loss and various home situations. The interviews were conducted in the local language, recorded, transcribed and analysed thematically. The findings show that nurses exchanged information with both healthcare (all cities) and social care services (Copenhagen, Tampere). Nurses in all cities, particularly Stockholm, reported to inform, and also convince patients to make use of home care. Nurses in Stockholm and Tampere reported that some patients refuse care due to co-payment. Nurses in these two cities were more likely to involve close relatives, possibly due to such costs. Not accepting care, due to costs, poses inequity in later life. Additionally, organisational changes towards a shift in location of care, i.e., from hospital to home, and from professional to informal caregivers, might be reflected in the work of the nurses through their initiatives to convince older patients to accept home care and to involve close relatives.

## 1. Introduction

The growing global trend for ageing populations has become a challenge to healthcare systems in the form of increased utilisation of healthcare services including hospital care [1]. The hospital discharge of older adults with both health and social care needs, known as complex care needs, is challenging as it involves multiple steps that require coordination between various teams in which nurses play a fundamental role [2]. Furthermore, the Nordic countries have seen a strong decline in the length of stay in hospitals in recent decades [3], which may imply greater care needs upon return home. Effective discharge planning procedures include collaboration with post-discharge services to ensure a smooth transition [4]. Clear responsibilities and goals facilitate the work of multidisciplinary discharge teams [5]. Nurses play a crucial role in assessing the patient’s needs and in the planning of resources and further assessments [6]. However, this is challenging as nurses are often exposed to heavy workloads with little time for preparing patients and their close relatives for the hospital discharge [7].

In the Nordic countries, hospital discharge planning often starts upon admission by nurses at the hospital. These nurses are also involved in the preparation of the discharge and the actual transition from hospital (typically back home). For older adults with complex care needs who return home, non-hospital-based nurses continue to provide post-discharge healthcare in their homes [8]. Across the Nordic countries, the concept ‘ageing in place’ has developed since it was introduced in the 1950s and refers to the idea that older people should rather remain living at home than moved into residential care homes. Subsequently, the number of beds in hospitals and, particularly, long-term care institutions have dramatically been reduced and the proportion of older adults who remain living at home has increased [9]. Larger numbers of older adults living at home have increased the pressure on home healthcare and social care services and family members, and have resulted in a higher share of older people seeking care at the hospital emergency. This is partly because the removal of institutional places has not been compensated for by a corresponding increase in home help services [10]. In recent years, it has been questioned whether ageing in place has become forced upon older adults and their families rather than an option [11]. This might be particularly relevant in Finland and Sweden where older adults have to apply for some care services themselves, negatively affecting care continuity post-hospital discharge [12]. Navigating the care systems and applying for care services can be demanding in older age due to, for instance, the digitalisation of processes in combination with cognitive decline, and tends to require support from younger relatives [13]. Nordic research studies have shown that it is common that nurses conduct work beyond their formal work tasks to compensate for the lack of close relatives as they often have strong feelings of responsibility for their patients [2]. In addition, the hospital discharge process has been shown to initiate communication between nurses and other care providers involved when staff are uncertain if information has been received and actions taken [8]. This shows that nurses compensate for gaps in the systems that could cause care inequity. Still, little is known about the work undertaken by nurses involved in the hospital discharge and post-hospital care of older adults with both healthcare and social care needs. Also, no study has compared such findings between welfare states where care services are primarily publicly funded through taxation, and most hospitals are publicly owned and managed [14,15]. Therefore, the aim of this study is to examine and compare what roles, responsibilities and actions nurses take in the hospital discharge process of older adults with complex care needs in three Nordic cities: Copenhagen (Denmark), Stockholm (Sweden) and Tampere (Finland).

## 2. Materials and Methods

### 2.1. Study Population and Participants

Copenhagen, Stockholm and Tampere were targeted as these cities are part of the research project Social Inequalities in Ageing (www.sia-project.se accessed on 1 August 2023).

Nurses in the three cities involved in any stage of the hospital discharge process including providing care to the older person upon their return home were eligible to participate. The job roles and workplaces of the 35 participating nurses are listed in Table 1. A total of 16 informants worked in Stockholm including 7 informants from 6 different general practices (of which 4 are located in disadvantaged areas), and 9 informants were recruited from geriatric departments at 7 different hospitals. Eleven informants worked in Copenhagen; four in home nursing care facilities in the municipality, two at the central health administration in the municipality and five in hospital departments. The majority of them worked in disadvantaged areas. The nurses employed in the central administration covered all areas of Copenhagen, and nurses from one of the selected home nursing care facilities also covered a more affluent area of the city. Eight informants worked in Tampere: three nurses from the city’s sole hospital with geriatric departments and five home care nurses across different areas of the city with no clear distinction between advantaged and disadvantaged areas.

### 2.2. Development of Vignettes

Vignette methodology seeks an understanding of people’s attitudes, perceptions and beliefs. It is considered a tool to prompt responses to interview questions, particularly useful when studying sensitive subjects such as healthcare [16]. For this study, three vignettes were developed by the researchers in consultation with medical professionals known to the research team (Appendix A). The vignettes represented older patients with age-related medical conditions of which one also had cognitive loss and one looked after their partner with dementia. Each vignette further described a scenario where concern about home healthcare, home help and the role of close relatives would be raised. This particularly included aspects of help received by children, environmental obstacles in their homes and the unwillingness of becoming a burden to the system. Professionals from all three countries with expertise in elderly care checked and provided feedback on the vignettes, which were refined. The professionals then confirmed that each vignette appeared useful in terms of its aim (face validity).

### 2.3. Recruitment and Data Collection

Recruitment was undertaken through the researchers’ professional networks, snowballing and invitation to departments with predominantly older patients such as geriatrics. Recruitment and data collection were undertaken in October–December 2018 in Copenhagen, December 2018–January 2019 in Tampere and January–April 2019 in Stockholm. The data collection in Stockholm was delayed to allow participants to report from the perspective of the new law-based routines on care coordination at hospital discharge introduced in autumn 2018. The vignettes were presented at the end of an interview on the same topic. Findings from the interview study have been published [8]. Each interview took about one hour, of which about 15–20 min was spent on the vignettes. This study mainly focused on the information provided by the participants about the vignettes; yet, information provided in the first part of the interview was occasionally included for contextualisation of data in cases when the informant referred back to their previous responses in the interview section. The interviews were undertaken face-to-face in the local language by one researcher in each city (AL, NKJ, JP—all female). All interviews were audio-recorded and transcribed verbatim. Written consent was obtained prior to data collection.

### 2.4. Data Analysis

A thematic analysis approach was used to analyse the data [17]. In accordance with thematic analysis, all transcripts were read by at least one native-speaking researcher of the local language (AL, JA, JP, NKJ) to familiarise with the data. Themes and sub-themes were identified and discussed in English between the researchers of which two are bilingual (JA: Danish and Swedish; JP: Finnish and Swedish). Words chosen for the themes and sub-themes were discussed to ensure that they reflected the local language. Descriptions of each theme and sub-theme were outlined and the codes were tested on one transcript in each language. The codes and descriptions were further discussed, agreed and refined into one codebook in English. The transcripts were then coded. At least two transcripts in each language were coded jointly by two researchers to ensure the coding of the remaining transcripts (undertaken by a native-speaking researcher of the local language) was applied in the same way for all transcripts. Quotes selected for the results were translated into English. The findings were discussed and interpretations were drawn collectively. 

### 2.5. Definitions of Home Care

In this study, home care refers to healthcare and/or social care. Social care includes receiving help to undertake activities in everyday life. In this study context, an older person with home care may for various reasons not receive both healthcare and social care but receives either healthcare or social care. In Copenhagen and Stockholm, home care refers to care in the person’s own home only. In Tampere, home care refers to health and social care in the person’s home as well as 24 h service housing in a care home that consists of small apartments for older adults. More detailed information on the organisation of care in these cities has previously been published [12].

## 3. Results

A total of 35 nurses were interviewed across the three cities: Copenhagen (*n* = 11), Stockholm (*n* = 16) and Tampere (*n* = 8). Four themes and nine sub-themes were identified (Table 2).

### 3.1. Care Considerations

#### 3.1.1. Nurses’ Passion for Caring and Spectrum of Actions to Meet the Patient’s Needs

Apart from undertaking their job tasks, all nurses from all cities expressed a genuine interest in providing the best possible care to meet the medical needs of patients in the situations described. This included, for example, ensuring that the patient receives their medication. A Finnish home care nurse described that determining the responsibility for providing and managing the patient’s medication is the first thing they do when they have a case like the vignette illustrating an older person who is discharged home after hospitalisation for a hip fracture. “*First, we need to find out who takes care of his medication. Has he done it himself [in the past], will family members take care of this, or are we going to help with this and how*”. (T9).

Danish and Swedish nurses also expressed that they consult other healthcare professionals at the hospital such as physiotherapists and occupational therapists as the health status of the patients often change over time, affecting the amount and type of care needed. Nurses in all cities also reported speaking directly to patients themselves or their close relatives to obtain a more comprehensive picture of the possible needs to undertake everyday life post-hospital discharge, including how to manage cleaning, cooking, moving in and outside of the house, etc. 

Some Danish and Swedish nurses thought that the care systems are challenging for older adults and reported on how they use their knowledge about the system to take action to address the patient’s needs in hospital and upon return home. “*I regard it as one of our major tasks, to act as a close relative and become the patient’s lawyer in the system*”. (C1).

The Swedish nurses referred to actions such as arranging for assessments and meetings that the patient has the right to, in order to visualise the patient’s care needs. “*[In such situation] I would like to arrange for a [care planning] meeting, no doubt, … that’s what I would fight for*”. (S13). This also included one Swedish informant reporting to occasionally offer and conduct cognitive tests at the hospital to add to the documentation of the older patient’s care needs and thus possibly speed up the process of getting a place in a care home.

#### 3.1.2. Information to and Dialogue with Involved Care Providers

The participating Danish nurses, irrespective of job location, reported regular contact between the hospital, the patient’s general practitioner and the home help service. They also demonstrated that they know exactly who to contact for certain information and assessments of the patient. “*I would directly look into nursing care options in this [case], because that’s what I am employed to do. Now, I’m not an initial assessor, so [first] I would ask the hospital [for the patient’s medical records]*”. (C10).

Both hospital and home care nurses in Tampere reported that they have contact with each other as well as with several other care providers to arrange for the help needed post discharge. Hospital and home care nurses discuss the patient’s situation at the hospital. They also discuss the patient’s everyday life at home on the phone and, during these phone calls, they set the discharge day, though this is not systemically carried out for every case. Finnish hospital nurses reported how doctors could express the need for home care after discharge (in the medical case summaries they send home with the patients), but as they are employed at the hospital, they cannot decide about home care. Nurses from home care said that they usually act according to the doctors’ suggestions, but sometimes wishes are set too high as is described below: “*…the doctor from the hospital has said ‘home care for two hours in the morning, one hour during the day, and two hours in the evening’. But that is not what we will do, especially as this client doesn’t have that many needs. Sometimes they [doctors at hospital] have too large hopes for what we can do in home care. But most of the time they don’t tell us how we should care for the clients, often there is a [written] comment that says ’at the beginning home care visits three times a day and [then] evaluate the need for future care’ […] and if agreed, we could visit three times a day for 1–2 weeks and then reassess their needs. From hospital it is difficult to evaluate the need for care at home*”. (T7).

The nurses in Stockholm explained how information is exchanged between healthcare professionals in the hospital and primary care nurses who provide home care using a digital system. Some of them reported that this included following up with the home help provider to ensure that there will be food and other essentials available upon the return home. If the patient has not had home help in the past, the nurses reported that they helped establish contact with the municipal home help manager. They also explained that they sometimes emphasise the patient’s needs when communicating with the home help manager: “*I am not allowed to recommend the patient a place at a care home, I’m not allowed to [do so] though I think that [a care home place] would be the best solution for the patient. I am only allowed to tell [the home help manager] that the patient needs certain help*”. (S12).

#### 3.1.3. Thoughts on Prevention of Infections and Avoidance of Hospitalisation

Most Danish nurses thought that infections could be prevented. *“It depends on what kind of infection it is, and it would include involving the patient’s general practitioner. It could be that there were some measures at home that could be taken to prevent the patient from getting this infection. It depends on whether it is a pneumonia or a urinary tract infection or what it is”.* (C1).

Whilst some Swedish nurses thought that certain hospitalisations could be avoided, others thought that it was difficult to prevent hospital admissions. “*It depends on what it [infection] is, whether it’s urinary tract infection or something else. I don’t think it’s possible to prevent any of the hospital admission that we have here*”. (S9).

Some Swedish nurses reported that they usually follow-up on a patient who is readmitted to hospital within 3 weeks, whereas others considered hospital readmission among older adults very common as this age group tends to have poor health status, resulting in an array of reasons for readmission. “*Well, readmission in 3 weeks… We currently have a patient who is frequently admitted to hospital, who has had a fall, wounds, pneumonia… Even if we have care planning meetings and try to prevent… it’s difficult. And this time of the year infections are common and so we do fairly little [preventative work]. I notice if someone has spent a lot of time in hospital and think that we probably must do something about it*”. (S2).

### 3.2. Nurses’ Communication with the Patient

#### 3.2.1. Understanding the Patient

Nurses in all cities reported having conversations with the patient in order to understand them and their individual needs. The nurses explained that they ask open-ended questions to gain an understanding of their needs and to be able to suggest suitable actions. “*We go through exactly what works fine and what doesn’t during a 24 h period. Some patients can’t do toileting themselves and need a lot of help. We suggest more resources to begin with and then cut down on resources as the patient’s ability to manage everyday life improves*”. (S14).

Contrary to the Swedish and Finnish nurses, some Danish nurses reported that patients can be offered a place in a short-term care home before returning home to better understand the patient’s needs: “*You could also consider a short-term care home for a short time so that you can assess how the patient actually functions 24 h a day, in order to provide the right help*”. (C4).

Understanding the patient also generated a more comprehensive picture and explanations to the patient’s decision and subsequent incidents, according to the Swedish nurses. 


*“[After having informed the patient] I ask ’What do you think?’ and they always reply ‘I want to go home’. Well, then they are sent home and occasionally it turns out they were discharged too early”.*
(S11)

#### 3.2.2. Anticipating Care Needs

Most Swedish nurses thought that following hospital discharge, it is particularly important to offer resources early in the recovery process and then possibly reduce the amount of care over time. Some of them also stressed that even though they believe that the patient would benefit from social care and therefore inform the patient about the possibility of social care services, they have to respect the patient’s decision whether to receive social care or not. Some Danish nurses had similar experiences including visiting the older patient upon their return home to present what services they offer. “*She [the patient] has said ‘you can come’, but she doesn’t want us [in her home], [she thinks] we are interfering. We can get such messages. And then we send someone out, usually the next morning, and then they carefully try to talk and tell and explain what we can offer, because the person may not be able to make a decision about it [home help] in the hospital because they are like this: ‘What’s that [home help] all about?’*”. (C10).

Finnish home care nurses reported that they value and take patients’ wishes into consideration, yet they also expressed their worries in cases where the current social care provided seemed insufficient. For example, regarding a case with a frail elderly man who had been readmitted to hospital, they would try to convince him to receive additional help even though he expressed no need for that. ”*He [the patient] must be exhausted from caring for his wife with dementia, so I would try to get them more home care visits or some kind of interval care [i.e., short-term stays in 24-h sheltered care] for the wife so that he could rest*”. (T8).

### 3.3. Individual Barriers to Care

#### 3.3.1. Language Barriers

Generally, nurses from all cities reported that professional interpreters are used when necessary. Some nurses in all cities further commented on using close relatives as interpreters. “*Then we bring an interpreter because relatives can sometimes translate a little differently, and maybe not quite what we say, so you have to be very careful there, [when] using relatives as interpreters*”. (C6).

In Stockholm, the nurses reported that professional interpreters were particularly used for individual care planning, whereas shorter conversations or check-ups might be translated by a colleague who speaks the same language as the patient, or a close relative. Homecare nurses who speak languages that are common in the area in which they work reported that, if possible, patients are allocated a nurse who speaks their mother tongue. “*We have staff speaking Arabic, Persian, simply just match [patient with staff]. … And it makes the patient feel safe, that homecare staff speak their language, not an interpreter unknown to them with no medical knowledge*”. (S3).

#### 3.3.2. Financial Barriers

Some Swedish and Finnish nurses provided examples of older patients who decline home help / home care for financial reasons as the patient has to contribute to such costs. “*Home help costs money and there are many [people] who do not want too much home help because they think it is too expensive and so they decline home help whereas home healthcare is free of charge*”. (S2).

Finnish nurses reported reminding older people that they may be eligible for monetary compensation for the home care costs, yet they experienced that older patients tend to decline the service offered anyway. “*…if older people need help and can’t pay for it, they can still get it, it is not because of money. However, these older people are often very determined, and if they don’t want any help, they won’t let nurses in their homes*”. (T9).

### 3.4. The Impact of Close Relatives

#### 3.4.1. Close Relatives’ Involvement in the Care

Nurses from Copenhagen and Stockholm thought that close relatives play an important role in supporting their older relative. Most Danish nurses reported that they consult close relatives as they may be able to provide useful information about the older person. “*It [the vignette] doesn’t say anything about relatives, but that would be number one: to get relatives’ perspective of how things are going at home*”. (C8).

The Finnish nurses described the role of close relatives as fluctuating and stressed that such a role presumes that there are close relatives and further depends on, e.g., how close they are and their possibilities to act as informal carers. The Finnish nurses emphasised that discussions with the patient and their relatives about their caregiving role influenced the amount of home care provided after hospital discharge. Whilst some Swedish nurses reported that it is the patient’s needs and own interest that guide the provision of social care including home help, others thought that close relatives were able to influence such decisions. Contrarily, all Finnish nurses said that relatives have no influences on the care decisions, as these are made solely based on the needs and wishes of the older person themselves with the exception of those with cognitive loss. “*in cases where the patient’s memory has declined, it is good to listen to relatives who knew the person when she was still able to express her own will, how she would like to be cared for. But, per se, it is always the client who decides*”. (T7).

Swedish and Finnish nurses often emphasised that close relatives should be asked whether they want to and are able to care for their older relative. Danish nurses thought that close relatives should primarily have enjoyable times together with their older relative rather than spending time on practical tasks. “*It’s great that the patient has got children who come over and they continue helping her with some tasks, yet, [they would] also [have the opportunity to] socialise more, if the patient gets municipal home help*”. (S5).

#### 3.4.2. Tensions with Close Relatives

Most nurses reported little or no experience of being threatened by close relatives. Nurses in all three cities had experience of close relatives requesting more care than decided, as illustrated in the third vignette with the demanding son of an older patient. For instance, some Swedish nurses reported having experienced situations with close relatives who demanded more home help or different types of social care (e.g., a place at a care home) than initially suggested by the social care manager. “*Then they [the social care manager] give in, that’s the kind of people the patient needs*”. (S10).

Nurses in all cities reported having handled situations with demanding close relatives of which some found the situations uncomfortable. Both Danish and Swedish nurses had witnessed how verbally aggressive close relatives had successfully secured more social care to their older relative. “*They [close relatives] pushed so much so that I got the impression that the managers were afraid of receiving a complaint*”. (C10).

Danish and Swedish nurses reported that when confronted by close relatives, they had explained that they are not in a position to make decisions on social care and referred the person to the municipality responsible for social care. Similarly, Finnish nurses reported that in such situations, they have discussed with the close relatives the reasons why more care would not be possible, referred to the care decision provided by the hospital, and offered to arrange for a meeting with the patient, close relatives, medical doctor and nurses from hospital and home care.

## 4. Discussion

Nurses in all three cities studied expressed a genuine interest in and reported undertaking numerous actions to provide the best possible care to meet older patients’ medical, but also social, care needs. Nurses explained how they get to know their patients through conversations, which helped them understand the patient’s care decisions, including the refusal of chargeable care services in Stockholm and Tampere. Finnish and Swedish nurses further reported asking close relatives whether they wanted to be involved in the social care to a greater extent than the Danish nurses who thought that close relatives should primarily socialise with their older relative. Particularly, Finnish nurses emphasised that care decisions including social care should be made solely based on the needs and wishes of the older person. 

### 4.1. Differences in the Nurses’ Views on Avoidable Hospital Readmissions

Nurses in all three cities expressed a genuine interest in providing the best possible care. This did, however, include different views on preventable hospitalisations, especially in Stockholm (see Section 3.1.3). Whilst most nurses thought that prevention of infection was possible, others considered readmission unavoidable. The vignette of a reoccurring infection that required hospital care did, however, not specify the type of infection. Hospital readmissions increase the risk of acquired hospital complications including infections [18] and are resource-intensive and expensive for the care system [19]. Potentially preventable hospitalisations have gained focus in recent decades [20]. Our study adds to such discussions by suggesting that nurses have different views on this, which might in turn impact the level of preventable hospitalisations. In a recent study on Australian healthcare professionals’ perceptions on readmission, the respondents thought that readmissions for, e.g., constipation and aspiration pneumonia, were often inevitable as patients admitted to hospital were often unable to undertake everyday self-care activities central to prevent complications and readmissions from occurring [21]. Subsequently, the Australian nurses questioned to what extent readmissions are avoidable. They also thought that hospital readmissions were difficult to address as they involve system-level factors and were considered as a low priority [22]. Across the three targeted Nordic cities, discharge planning, transitional care and comprehensive geriatric assessment are examples of hospital interventions implemented [8] and internationally shown to be efficient in the prevention of readmissions [22,23]. Yet, the Australian study concluded that healthcare professionals’ lack of awareness and prioritisation of avoiding readmissions resulted in multiple inherent contradictions in the findings with no single or immediate solution [23]. Individual variations in awareness may reflect the mixed findings on hospital readmission in this study, too. 

### 4.2. Information Exchange Seems Facilitated by Jointly Organised Health and Social Care

In terms of information exchange and dialogue with the involved care providers, Danish and Finnish nurses reported making contact with home care services (see Section 3.1.2), which provide both healthcare and social care and are organised jointly in both Copenhagen and Tampere [11]. Contrarily, Swedish nurses reported to mainly exchange information within the regional healthcare services and reported having very little contact with social care, which, in Stockholm, is organised by the municipalities. Older adults with multimorbidity often depend on both healthcare and social care services and the lack of integration between the two particularly negatively affects vulnerable groups such as older adults. According to Swedish law (2017:612) that came into practice in 2018 yet not evaluated, an individual care coordination plan should be developed in collaboration with both healthcare and social care actors and the patient to address complex care needs. Patients should also be allocated a healthcare professional in primary care as their main contact person; however, in 2020, this was only the case for about half of the patients [24]. In Copenhagen and Tampere, information exchange between healthcare and social care does not seem to be a problem of such magnitude, possibly because healthcare and social care are organised at the same administrative level, which may facilitate care coordination [11]. Additionally, previous research has shown that information exchange is also challenged by limited access to patient records between healthcare and social care services in Copenhagen and Stockholm, and a lack of electronic communication channels between hospital and municipality staff in Tampere [8].

### 4.3. Challenges in Co-Payment Reported in Personal Conversations

In their conversations with the older patients, nurses had experienced that individuals expressed their concerns in affording out-of-pocket costs for care in Sweden (social care at home) and, particularly, Finland (health and social home care) (see Section 3.3.2). The Finnish nurses, however, also reported that some older adults are very determined in not receiving care. Refusing home care might threaten the older person’s health and safety as not receiving timely care can worsen health conditions and lead to poorer health outcomes [25]. A recent study has shown that relatively few older Europeans postpone or skip medical care because of costs. This study provided information on Sweden (but not Denmark and Finland), which showed that only 1% of Swedish adults aged 65 years and over had skipped medical treatment or not consulted a doctor when having a medical problem, because of costs [26]. Nonetheless, in terms of social care, nurses reported that some older adults have told them that they minimise the amount of social care or refuse help, for financial reasons [26]. This could be because of the lack of knowledge of how to navigate the system and apply for reimbursement. It has also been speculated that older adults who have a pension just above the threshold to receive compensation for care expenditures, and thus do not qualify for reimbursement, are the ones who refuse chargeable care [26]. Other potential reasons include older people not wanting to feel burdensome or accept loss of independence. Furthermore, differences between the cities in involving close relatives might reflect the co-payment systems. The findings show that nurses in Stockholm and Tampere more often emphasised that close relatives should be asked whether they want or can care for their older relative, compared to nurses in Copenhagen (see Section 3.4.1). It could be that the involvement of close relatives was not brought up by Danish nurses, as both healthcare and social care are fully tax-funded in Copenhagen compared to Stockholm and Tampere where patients have to contribute to such costs. This raises concern about the provision of informal care and inequity as some older adults have no informal caregiver. Nurses’ initiatives to convince older patients to accept home care and to involve close relatives also add to research on the “ageing in place” policy, which refers to changes in the organisation of care services in the Nordic countries towards a shift in location of care, i.e., from hospital to home, and a shift from professional to informal caregivers [27]. Further research is needed to examine the role, experiences and perceptions of informal caregivers in the care systems for older adults, a research field less studied in the Nordic countries [28].

### 4.4. Study Strengths and Limitations

Study strengths include the use of vignettes, i.e., presenting the same scenarios for all participants, allowing for cross-country comparisons, though comparison to countries worldwide is limited. Also, the vignettes had been face-validated by non-participating experts in the research field. Researchers developed and refined the coding system together and participating nurses were involved in different parts of the hospital discharge including post-hospital care. Limitations include that the nurses were not observed but asked about their actions in certain situations. However, vignettes are particularly useful to obtain precise answers on a topic that is considered sensitive [17]. Further, the views of other professionals, older adults and their close relatives were not captured, providing a single-sided perspective. Furthermore, the results from the interview and the vignettes were presented separately. Also, this study was conducted in two larger cities and one smaller city. Whilst this might limit the comparability of the cities, the use of the same vignettes is likely to have generated data that reflect the research topics rather than differences in the data collection. 

## 5. Conclusions

This study has identified multiple differences and similarities in the roles and experiences of nurses involved in the hospital discharge of older adults in Copenhagen, Stockholm and Tampere. The separate organisations of healthcare and social care services in Stockholm resulted in Swedish nurses reporting little information exchange with social care services. A different organisational structure may facilitate information exchange, as seen in Denmark and Finland. Nurses in Stockholm and Tampere, where co-payment is applied for certain care, reported that some of their patients refuse care due to the costs. Compared to nurses in Copenhagen, nurses in Stockholm and Tampere reported that some of their patients refuse care due to the costs and that they were more likely to ask close relatives about their potential involvement in social care, possibly to find alternatives to the co-payment systems of these cities. These initiatives by the nurses may reflect the influences of the “ageing in place” policy and bring attention to the role of close relatives. Further research is needed to examine informal caregivers’ perceptions on their roles in the Nordic care systems for older adults. 

## Figures and Tables

**Table 1 ijerph-20-06809-t001:** Nurses’ job roles and locations.

Informant	Job Role	Workplace
Copenhagen		
C1	Home healthcare nurse	Home nursing care
C2	Home healthcare nurse	Home nursing care
C3	Discharge coordinator (nurse)	Municipality of Copenhagen
C4	Discharge coordinator (nurse)	Municipality of Copenhagen
C5	Nurse	Medical department at hospital
C6	Nurse	Medical department at hospital
C7	Follow home nurse	Hospital (several departments)
C8	Coordination consultant (nurse)	Hospital (several departments)
C9	Coordination consultant (nurse)	Hospital (several departments)
C10	Initial assessor (nurse)	Home nursing care
C11	Initial assessor (nurse)	Home nursing care
Stockholm		
S1	Home healthcare nurse	General practice
S2	Home healthcare nurse	General practice
S3	Home healthcare nurse	General practice
S4	Home healthcare nurse	General practice
S5	Home healthcare nurse	General practice
S6	Home healthcare nurse	General practice
S7	Home healthcare nurse	General practice
S8	Nurse with care coordination responsibilities	Geriatric department at hospital
S9	Nurse with care coordination responsibilities	Geriatric department at hospital
S10	Nurse with care coordination responsibilities	Geriatric department at hospital
S11	Nurse with care coordination responsibilities	Geriatric department at hospital
S12	Nurse with care coordination responsibilities	Geriatric department at hospital
S13	Nurse with care coordination responsibilities	Geriatric department at hospital
S14	Care coordinator (qualified nurse)	Geriatric department at hospital
S15	Care coordinator (assistant nurse)	Geriatric department at hospital
S16	Care coordinator (assistant nurse)	Geriatric department at hospital
Tampere		
T1	Nurse with discharging responsibilities	Geriatric department at hospital
T2	Nurse with discharging responsibilities	Geriatric department at hospital
T3	Nurse with discharging responsibilities	Geriatric department at hospital
T4	Nurse with discharging and care continuity responsibilities	Home care
T5	Nurse with discharging responsibilities	Discharging team
T6	Nurse with discharging and care continuity responsibilities	Home care
T7	Nurse with discharging and care continuity responsibilities	Home care
T8	Nurse with discharging and care continuity responsibilities	Home care

**Table 2 ijerph-20-06809-t002:** Themes and sub-themes.

Themes	Sub-Themes
Care considerations	Nurses’ passion for caring and spectrum of actions to meet the patient’s needs
	Information to and dialogue with involved care providers Thoughts on prevention of infections and avoidance of hospitalisation
Nurses’ communication with the patient	Understanding the patient Anticipating care needs
Individual barriers to care	Language barriers Financial barriers
The impact of close relatives	Close relatives’ involvement in the care Tensions with close relatives

## Data Availability

The data presented in this study are available on request from the corresponding author in the local languages with all identifiable information removed. The data are not publicly available, due to their containing information that could compromise the privacy of the participants.

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
