# Peer review of "Nurses’ Roles, Responsibilities and Actions in the Hospital Discharge Process of Older Adults with Health and Social Care Needs in Three Nordic Cities: A Vignette Study"

_ijerph, 2023, doi:10.3390/ijerph20196809_

Round 1
Reviewer 1 Report
This is a well written paper presenting part of the findings from a qualitative study exploring nurses' roles and responsibilities in identifying discharge care needs of patients with complex care from hospital. I read the article carefully and have some comments for improvement.
Introduction: In line 46 a small editing error, instead of "...continue providing" change to "continue to provide"
It would be good to add a sentence or two to explain what is meant by complex care needs, what this term involves.
It would be good to justify better the comparison of the three Nordic countries. The authors report that their welfare systems are different, but how different are they to deem such comparison?
Materials: In line 107, apart from checking the vignettes, what process was followed between the reviewers to confirm face validity?
Recruitment: It is unclear why the authors decided to present the interview data separate to the vignette data when the aim of the study was to explore the role and responsibilities of the nurses. I think the paper would benefit from presented the full dataset and the richness of the data from the using different methods and improve triangulation.
Data analysis: The authors report that they present some of the data from the previous paper they have published on this study, but it is not clear where the distinction is made between data from the vignettes and data from the previous paper.
Development of codebook: Was a separate codebook developed in each country of was one codebook from all three countries?
How did authors ensure that codes and themes developed from 3 different languages had the same meaning? In what language was the final codebook developed? The themes that are presented in table 2 are themes developed only from this vignette study or are they included in the previous interview study?
Discussion: The themes are well explained. However, there is limited link to the headings and themes discussed in the discussion section. It would be advised to demonstrate how the themes in the results feed into the discussion.
Conclusion: There is a lot of repetition of the results. I would suggest to conclude with some recommendations for service change and development, and for future research in the area of discharge and liaison with the community services. Also, please add to the limitation the separate presentation of the results (i.e. the vignette data and the interview data).
Author Response
We thank the reviewers for their comments. Please see our point-by-point responses below.
Reviewer 1
This is a well written paper presenting part of the findings from a qualitative study exploring nurses' roles and responsibilities in identifying discharge care needs of patients with complex care from hospital. I read the article carefully and have some comments for improvement.
Introduction: In line 46 a small editing error, instead of "...continue providing" change to "continue to provide"
Response: We have now changed the wording into “continue to provide”. (page 2, lines 48-9)
It would be good to add a sentence or two to explain what is meant by complex care needs, what this term involves.
Response: Good suggestion. We have now specified that complex care refers to having both health care needs and social care needs. “Hospital discharge of older adults with both health and social care needs, known as complex care needs, is…” (page 1, lines 35-36)
It would be good to justify better the comparison of the three Nordic countries. The authors report that their welfare systems are different, but how different are they to deem such comparison?
Response: Thank you commenting on this. We have now removed the word ‘different’ and specified similarities in these welfare systems and added references to literature in which comparison of the Nordic countries is presented and discussed. “between welfare states where care services primarily are publicly funded through taxation and most hospitals are publicly owned and managed.” (page 2, lines 75-76)
Materials: In line 107, apart from checking the vignettes, what process was followed between the reviewers to confirm face validity?
Response: We have now specified that the experts “checked and provided feedback on the vignettes which were refined.” (page 5, lines 111-112)
Recruitment: It is unclear why the authors decided to present the interview data separate to the vignette data when the aim of the study was to explore the role and responsibilities of the nurses. I think the paper would benefit from presented the full dataset and the richness of the data from the using different methods and improve triangulation.
Response: The data from the interviews generated perspectives on socioeconomic status and close relatives that were not discussed to such extent in the vignettes. The vignette data were more concentrated to the nurses’ roles, responsibilities and actions.
Data analysis: The authors report that they present some of the data from the previous paper they have published on this study, but it is not clear where the distinction is made between data from the vignettes and data from the previous paper.
Response: In some interviews, when commenting on the vignettes, the interviewee referred back to their previous responses in the interview section. In these cases, the responses and information provided in the interview section were used. We have now added “information provided in the first part of the interview has occasionally been included for contextualisation of data in cases when the informant refereed back to their previous responses in the interview section.” (page 5, lines 126-127)
Development of codebook: Was a separate codebook developed in each country of was one codebook from all three countries?
Response: We have now specified that “The codes and descriptions were further discussed, agreed and refined into one codebook in English.” (page 5, line 140)
How did authors ensure that codes and themes developed from 3 different languages had the same meaning? In what language was the final codebook developed? The themes that are presented in table 2 are themes developed only from this vignette study or are they included in the previous interview study?
Response: We have now clarified this: “Themes and sub-themes were identified and discussed in English between the researchers of which two are bilingual (JA: Danish and Swedish; JP: Finnish and Swedish). Words chosen for the themes and sub-themes were discussed to ensure that they reflected the local language.” (page 5, lines 135-137)
Discussion: The themes are well explained. However, there is limited link to the headings and themes discussed in the discussion section. It would be advised to demonstrate how the themes in the results feed into the discussion.
Response: In the Discussion we now refer to specific sections in the Results. (pages 12-13)
Conclusion: There is a lot of repetition of the results. I would suggest to conclude with some recommendations for service change and development, and for future research in the area of discharge and liaison with the community services. Also, please add to the limitation the separate presentation of the results (i.e. the vignette data and the interview data).
Response: We have now rewritten the Conclusions and added to the limitations as suggested. (page 14)
Reviewer 2 Report
I find this paper interesting and highly topical.
It is not clear to me the definition of home care, I think section 2.5 should be clarified both in conceptual and language terms.
The main question addressed by the research is how nurses experience the ordinary discharge of elderly patients who need further assistance after hospitalisation, how they use to organize it and what challenges do they face.
The topic is relevant to the field and adds empirical considerations which are very useful both in qualitative research terms and case report.
I have not read similar papers by now, that's why I consider their approach interesting, although I can certainly have missed similar contributions.
I think the study design is appropriate for how it is conceived.
The conclusions are consistent with the evidence and arguments presented and they do address the main question posed.
The references are appropriate. I have further references about healthcare fragmentation and discharge issues among elderly patients but they are all papers in Italian. Instead, the following one considers the problem in a more international perspective considering the utility of telemedicine and nursing case management, and I recommend to add it in the background:
Pennestrì F, Banfi G. Primary Care of the (Near) Future: Exploring the Contribution of Digitalization and Remote Care Technologies through a Case Study. Healthcare (Basel). 2023 Jul 27;11(15):2147. doi: 10.3390/healthcare11152147. PMID: 37570387; PMCID: PMC10418748.
I have no specific comments on the tables.
Author Response
We thank the reviewers for their comments. Please see our point-by-point responses below.
Reviewer 2
I find this paper interesting and highly topical.
It is not clear to me the definition of home care, I think section 2.5 should be clarified both in conceptual and language terms.
Response: We thank the reviewer for this suggestion. We have now rewritten this paragraph. (page 5, lines 147-156)
The main question addressed by the research is how nurses experience the ordinary discharge of elderly patients who need further assistance after hospitalisation, how they use to organize it and what challenges do they face.
The topic is relevant to the field and adds empirical considerations which are very useful both in qualitative research terms and case report.
I have not read similar papers by now, that's why I consider their approach interesting, although I can certainly have missed similar contributions.
I think the study design is appropriate for how it is conceived.
The conclusions are consistent with the evidence and arguments presented and they do address the main question posed.
The references are appropriate. I have further references about healthcare fragmentation and discharge issues among elderly patients but they are all papers in Italian. Instead, the following one considers the problem in a more international perspective considering the utility of telemedicine and nursing case management, and I recommend to add it in the background:
Pennestrì F, Banfi G. Primary Care of the (Near) Future: Exploring the Contribution of Digitalization and Remote Care Technologies through a Case Study. Healthcare (Basel). 2023 Jul 27;11(15):2147. doi: 10.3390/healthcare11152147. PMID: 37570387; PMCID: PMC10418748.
Response: We have read the suggested paper but cannot see how it adds to neither the Background nor the Discussion of our study.
I have no specific comments on the tables.
Reviewer 3 Report
1. The introduction was focused and ended with the aim. This was a qualitative study with interviews and a small sample and used developed vignettes.
2. The format of the study was presented.
3. Themes were presented and quotes from nurses about the theme were shared.
4. The visits to the home and costs are country specific in terms of insurance and what is allowed. The limitation section referenced the issue of country but is limited in comparison of countries worldwide.
5. The discussion started with a focus on readmissions, but that was not clear in the themes above. Other concepts were presented in the discussion with references to literature.
6. In the conclusions, what additional research with this focus should be done?
7. The authors stated consent was obtained.
8. Tables are appropriate and add to the paper.
Author Response
We thank the reviewers for their comments. Please see our point-by-point responses below.
Reviewer 3
- The introduction was focused and ended with the aim. This was a qualitative study with interviews and a small sample and used developed vignettes.
- The format of the study was presented.
- Themes were presented and quotes from nurses about the theme were shared.
- The visits to the home and costs are country specific in terms of insurance and what is allowed. The limitation section referenced the issue of country but is limited in comparison of countries worldwide.
Response: Thank you for this suggestion. We have now added that comparison to countries worldwide is limited. (page 13, lines 452-453)
- The discussion started with a focus on readmissions, but that was not clear in the themes above.
Other concepts were presented in the discussion with references to literature.
Response: In the Discussion, we now refer to specific Results sections. (pages 12-14)
- In the conclusions, what additional research with this focus should be done?
Response: We think that this research field would benefit from studying the views of informal caregivers who provide social care to older adults. (page 14, lines 486-487)
- The authors stated consent was obtained.
- Tables are appropriate and add to the paper.
Reviewer 4 Report
The paper under the title "Nurses’ Roles, Responsibilities, and Actions in the Hospital 2 Discharge Process of Older Adults with Complex Care Needs 3 in three Nordic Cities: An Interview Study" is written well with provides clarity in thoughts organization. The title can be improved. The introduction is also written well. However, I suggest that the authors add a separate section for the literature review which must describe the Nurses’ Roles, Responsibilities, and Actions similar you presented in the discussion of this paper. It will provide a more clear understanding to the general readers. The rest of the paper sections are very well written.
Best Wishes
Author Response
We thank the reviewers for their comments. Please see our point-by-point responses below.
Reviewer 4
The paper under the title "Nurses’ Roles, Responsibilities, and Actions in the Hospital 2 Discharge Process of Older Adults with Complex Care Needs 3 in three Nordic Cities: An Interview Study" is written well with provides clarity in thoughts organization. The title can be improved.
Response: Thank you for this suggestion. We have now replaced ‘complex care’ with ‘health and social care’. We have also replaced ‘An Interview Study’ with “A Vignette Study’ (page 1, lines 3-4)
The introduction is also written well. However, I suggest that the authors add a separate section for the literature review which must describe the Nurses’ Roles, Responsibilities, and Actions similar you presented in the discussion of this paper. It will provide a more clear understanding to the general readers.
Response: We have now added that “Nurses play a crucial role in assessing the patient’s needs and in the planning of resources and further assessments.” (page 1, lines 42-43)
The rest of the paper sections are very well written.